# Well-Being of Healthcare Workers and the General Public during the COVID-19 Pandemic in Vietnam: An Online Survey

**DOI:** 10.3390/ijerph18094737

**Published:** 2021-04-29

**Authors:** Tuyen Dinh Hoang, Robert Colebunders, Joseph Nelson Siewe Fodjo, Nhan Phuc Thanh Nguyen, Trung Dinh Tran, Thang Van Vo

**Affiliations:** 1Institute for Community Health Research, University of Medicine and Pharmacy, Hue University, Hue 530000, Vietnam; tuyenhoang@hueuni.edu.vn (T.D.H.); nptnhan@huemed-univ.edu.vn (N.P.T.N.); 2Faculty of Public Health, University of Medicine and Pharmacy, Hue University, Hue 530000, Vietnam; 3Global Health Institute, University of Antwerp, 2000 Antwerp, Belgium; robert.colebunders@uantwerpen.be (R.C.); JosephNelson.SieweFodjo@uantwerpen.be (J.N.S.F.); 4Faculty of Public Health, Da Nang University of Medical Technology and Pharmacy, Da Nang 550000, Vietnam; trandinhtrung@dhktyduocdn.edu.vn

**Keywords:** WHO-5 well-being, COVID-19, social distancing, preventive measures, Vietnam

## Abstract

The COVID-19 pandemic and associated restrictive measures implemented may considerably affect people’s lives. This study aimed to assess the well-being of Vietnamese people after COVID-19 lockdown measures were lifted and life gradually returned to normal. An online survey was organized from 21 to 25 April 2020 among Vietnamese residents aged 18 and over. The survey was launched by the Hue University of Medicine and Pharmacy. The WHO-5 Well-Being Index (scored 0–25) was used to score participants’ well-being. A multivariate logistic regression model was used to determine the predictors of well-being. A total of 1922 responses were analyzed (mean age: 31 years; 30.5% male; 88.2% health professionals or students in the health sector). The mean well-being score was 17.35 ± 4.97. Determinants of a high well-being score (≥13) included older age, eating healthy food, practicing physical exercise, working from home, and adhering to the COVID-19 preventive measures. Female participants, persons worried about their relatives’ health, and smokers were more likely to have a low well-being score. In conclusion, after the lockdown measures were lifted, the Vietnamese have people continued to follow COVID-19 preventive measures, and most of them scored high on the well-being scale. Waiting to achieve large-scale COVID-19 vaccine coverage, promoting preventive COVID-19 measures remains important, together with strategies to guarantee the well-being of the Vietnamese people.

## 1. Introduction

On 31 December 2019, unexplained cases of pneumonia were reported in Wuhan City, Hubei Province, China by the World Health Organization (WHO) China Country Office [1]. Three months later, the WHO officially declared the Coronavirus disease 2019 (COVID-19) outbreak as a pandemic, with more than 118,000 confirmed cases and over 4200 death cases [2]. Prior to the identification of the first COVID-19 case in Vietnam, the Vietnamese government proactively took measures to prevent the importation of the disease into the country. Health screening was organized at the country entry points, and people were advised to practice personal hygiene and to wear a face mask in public places. Initially, around the end of January 2020, the COVID-19 cases were imported into Vietnam from China. In early March 2020, cases entered from Europe and America [3]. Faced with an increasing number of COVID-19 cases, from 1 April to 22 April 2020, the Vietnamese government implemented a nationwide lockdown alongside other preventive measures such as keeping a minimum distance of 2 m from others, staying at home, wearing a face mask, washing hands regularly, and a restriction of gatherings. People from abroad who entered Vietnam were subject to compulsory isolation for 14 days [4]. Thanks to these measures, COVID-19 transmission stopped until 25 July 2020, when new cases of COVID-19 infection appeared, and local community transmission was detected in the city of Da Nang. This second wave of COVID-19 in Vietnam was successfully controlled, but a third wave appeared in Hai Duong Province on 27 January 2021. Additionally, this third wave was rapidly controlled. In total, only 35 COVID-19-related deaths have been reported in Vietnam (Table 1). 

Currently, as there are no new domestic cases of COVID-19 in Vietnam, the country has applied a “new normal” state with the partial relaxation of social distancing measures, due to the concern that a complete removal of these measures could lead to a resurgence of the pandemic, as observed in other countries [8]. On 8 March 2021, COVID-19 vaccination started in Vietnam with the AstraZeneca vaccine among frontline healthcare workers, followed by essential service providers, teachers, people with chronic diseases, and people living in epidemic areas. Nationwide vaccination is planned with the locally produced vaccine Nanocovax by the end of 2021. 

Worldwide, the COVID-19 pandemic and the associated preventive measures have had a major effect on people’s lives, causing anxiety and stress, affecting daily life activities at home and at workplaces, and restricting social relationships [9]. To investigate whether people were adhering to the preventive measures implemented by the Vietnamese government, the first online survey was organized 31 March–6 April 2020 when the lockdown measures were still in place. The results of this initial survey showed good preventive behavior of the Vietnamese population [10]. The aim of the second online survey was to investigate the effect of the COVID-19 pandemic on the COVID-19 preventive behaviors and the well-being of the Vietnamese people when the lockdown measures were relaxed starting 23 April 2020 and life gradually returned to normal in Vietnam.

## 2. Materials and Methods

### 2.1. Study Design

Cross-sectional study collecting data through an online survey in Vietnam from 21 to 25 April 2020.

### 2.2. Study Procedures

Data of the study were collected via a web-based online survey tool developed by the ICPcovid consortium (https://www.icpcovid.com/) (accessed on 26 April 2020). The survey was launched by the Institute for Community Health Research, Hue University of Medicine and Pharmacy, Vietnam. The website interface was designed to be easily accessible by various devices, such as computers, tablets, and smart phones. The entire questionnaire could be filled out in 10 min or less and was totally anonymous (no identification information was collected). Each question needed to be answered before the participant could go to the next question. Eligible participants were Vietnamese aged 18 years or older who were able to read and understand Vietnamese and resided in Vietnam at the time of data collection. Snow-ball sampling was used to recruit the participants. The survey link was shared via various social media platforms to relatives, friends, and colleagues. 

Different determinants of well-being were collected (sociodemographic characteristics, health status and determinants of health, adherence to preventive measures, and consequences of the preventive measures) (Figure 1). The level of anxiety about the health of the participant and his/her relatives was measured by a 5-point Likert scale (1 = not worried/afraid to 5 = extremely worried/afraid); a score of ≥3 was considered as a moderate-to-high level of anxiety. Twenty yes/no questions were asked to assess the participant’s adherence to the COVID-19 prevention measures, including 9 questions regarding the adherence to preventive measures at the personal level and 11 questions regarding the adherence to preventive measures at the community level. Data were compared with data of the first survey using the same adherence scale [10].

Well-being was scored using the five questions recommended by WHO: “I have felt cheerful in good spirits”, “I have felt calm and relaxed”, “I have felt active and vigorous”, “I woke up feeling fresh and rested”, and “My daily life has been filled with things that interest me”. Each answer was rated on a scale range from 0 to 5. The overall score ranged from 0 to 25, with 0 representing the worst probable and 25 representing the best probable well-being. A score below 13 indicated poor well-being (WHO 1998) [11].

### 2.3. Data Analysis

IBM (Armonk, NY, USA) SPSS version 20 was used to analyze the collected data. Continuous variables were described by means and standard deviations (SD). Categorical variables were described by frequency (*n*) and percentage (%). Well-being was the dependent variable. A multiple logistic regression model was used to analyze which independent variables were predictors of poor well-being (score < 13). First, age and gender were included in the model to adjust other covariates. Later, all factors available in the conceptual framework (Figure 1) were kept in the model if they had a statistically significant relationship with the dependent variables at a *p*-value < 0.05. 

## 3. Results

### 3.1. Characteristics of Study Participants

Of the 1934 responses obtained during the survey, 1922 were eligible for analysis. The respondents lived in 46 of the 63 provinces and municipalities of Vietnam, with more than half residing in urban areas. The average age was 31 years (SD: 10; range: 18–76 years). One thousand three hundred and seventy-six (71.6%) of the respondents reported living with children. Among the 332 (17.3%) respondents who were at least 40 years old, 27 (8.1%) were at least 60 years old (Table 2).

The education level was higher in male than female participants, but the rate of unemployment was lower in females than males (Table 3). 

### 3.2. Consequences of the COVID-19 Pandemic on People’s Lives 

Four hundred and six participants (21.1%) were moderately or very worried about their own health and 517 (26.9%) about the health of their relatives (Table 4). Ninety-seven (5.0%) reported difficulties in obtaining food. Of the 135 people with an underlying disease, nine (6.7%) encountered difficulties in obtaining medication (Table 4). Nearly 90% of participants were physically active during the pandemic, and 74.4% of them practiced outdoor activities. About 80% of the 1376 participants who lived with children responded that they participated in activities with their children on a daily basis.

### 3.3. COVID-19 Preventive Behavior among Respondents

The adherence to personal preventive measures remained high during the second survey, with rates ranging from 55.9% to 99.9%. Only a temperature check at least twice a week and disinfecting one’s phone were seldom reported. (Table 5).

Adherence to community preventive measures also remained high during the second survey, with rates ranging from 43.9% to 99.7%, but most people continued going regularly to a market (Table 6). 

### 3.4. Well-Being during the COVID-19 Pandemic

Three hundred and ten (16.1%) persons had a poor well-being score (overall well-being score less than 13). The mean scores for each item on the WHO well-being scale are summarized in Table 7.

The multivariable model found that the following factors were associated with reduced odds for poor well-being (predictors of a well-being score ≥13): male gender, eating more healthy food, physical activity, working from home, and adherence to the COVID-19 preventive measures. In contrast, an age below 40 years, being worried about their relatives’ health, and being a smoker were all associated with poor well-being (Table 8).

There was no difference about poor well-being between, during, and after social distancing based on the participation’s response date.

A similar proportion of poor well-being was reported by those who responded on April 21 to 22 (lockdown period) (16.4%) and those who responded April 23–25 (when the lockdown measures were relaxed) (15.6%) (Table 9).

## 4. Discussion

Despite the COVID-19 pandemic and the stringent restrictive measures that had been implemented in Vietnam, relatively few participants (16.1%) scored low on the WHO well-being score. This figure was lower than in Wuhan, China, where 48.3% respondents scored low using the same scale [12]. In our study, the mean score of the five components was 17.35 ± 4.97, and this was higher compared to Austria after four weeks of lockdown (15.05 ± 5.40) [13]. A similar study during a period of COVID-19 lockdown in the UK also found a lower mean WHO well-being score (10.43 ± 5.40) [14]. The relatively moderate impact of the COVID-19 epidemic and the implemented preventive measures on the well-being of the Vietnamese may be because the epidemic in Vietnam was rapidly controlled, and the population accepted adhering to the preventive measures. 

The impacts of the COVID-19 preventive measures on the physical and mental well-being have been documented by many studies. Long-term adherence to these measures, as well as negative information about the epidemic, may affect the physical and mental well-being in the population [15]. Studies evaluating mental health during the lockdown periods in Austria and the US showed that young people, women, the unemployed, and low-income people seemed to be more stressed than others [13,16]. In another online survey in Vietnam during the lockdown period, females, having chronic diseases, and having a high number of family members was associated with poor well-being and a lower quality of life [17]. Certain participants in our study reported difficulties in obtaining food or medication, feeling worried about their own health or that of their relatives, and a few experienced violence or discrimination. These experiences most likely affected their well-being. The WHO Department of Mental Health and Substance Use has provided advice to improve mental and psychosocial well-being during the pandemic [18]. People in the community should sympathize and help those affected by the pandemic, avoid discrimination against infected people, and follow official COVID-19 information from local health authorities. Indeed, uncontrolled infodemics could make people feel anxious or distressed. In addition, healthcare workers should consider mental and psychosocial well-being as important as physical health. 

Our multivariable analysis showed that better well-being was associated with eating healthy food, practicing physical exercise, and observing the COVID-19 preventive measures. Notably, increasing age was associated with better well-being (Table 8), suggesting that the pandemic may be particularly detrimental to the well-being of younger individuals who may find it more difficult to endure confinement compared to older persons. Similar findings were recently reported in the USA, where poor mental health and well-being during the COVID-19 pandemic was reported among persons below the age of 40 years [16]. Other studies have shown that persons above the age of 60 years may be more worried about the risk of disease or death and may experience reduced well-being as a consequence of confinement [19,20]. In our survey, only 27 respondents were at least 60 years old, making this group too small for proper investigation in the multivariable model. Moreover, the successful interruption of COVID-19 transmission in Vietnam at an early stage of the pandemic and the absence of reported deaths due to COVID-19 during the first wave may have put the population older than 40 years at greater ease [10]. In addition, the risk of losing a job or decreased income during the pandemic might have been lower in persons above the age of 40 years as compared to younger people [21].

Female gender was also associated with poorer well-being in our study. This is similar to previous findings from Austria, Denmark, the UK, and Vietnam [13,17,22,23]. An explanation could be that women carry the double burden of having a job and family responsibilities [24]. Additionally, a recent study of six countries in Europe, North America, and Asia showed that the impact of the COVID-19 pandemic led to gender inequality in many different aspects. Women had a higher rate of permanent job loss than men, and their income from labor decreased more than that of men [25].

Fear and worry about their relatives’ health were associated with poor well-being. This reflects the concerns that respondents have for their loved ones, as they do not want them to develop COVID-19. Indeed, SARS-CoV-2 may spread rapidly among family clusters [26]. Understandably, being a smoker was also associated with poor well-being, since some of the risk factors that increase the severity of COVID-19 disease (lung disease, cardiovascular disorders, and diabetes) are more common among smokers. Therefore, quitting smoking is recommended, especially for those with underlying diseases [27]. The finding that physical activity was associated with a higher well-being score resonates with previous studies, which found that physical activity improves mental well-being, in addition to reducing the risk of acute respiratory distress syndrome, which is a major cause of death in COVID-19 patients [28].

In many countries, the COVID-19 pandemic resulted in reduced income and increased food prices. Food insecurity and difficulties in accessing healthy food may lead to malnutrition and mental health problems [29]. However, Vietnam has a policy of controlling food prices and guaranteeing food security by a well-organized collaboration between the government, producers, and supermarkets [30]. This explains why only 5% of our respondents reported difficulties in obtaining food and that nearly 90% responded that they were regularly eating more healthy food during the outbreak. 

Some limitations of the study should be mentioned. The key disadvantage of our snowball sampling-based online survey included a community bias. Indeed, it is possible that the first participants determined the remaining participants, as they shared the link only to their networks. This was partially mitigated by sharing the survey link on several different platforms. The survey was launched in medical schools, which resulted in a large percentage of respondents being medical students and healthcare workers. Such a population is likely to be more aware and compliant with preventive measures. The nonrandom selection of participants, the selection bias of the online approach, and the adoption of a convenient sample size rendered our study population non-representative of the general population in Vietnam. Moreover, in an online questionnaire, there is a risk for recall bias and/or the submission of incorrect information by respondents.

## 5. Conclusions

Thanks to the strict preventive measures that were implemented in Vietnam and the excellent preventive behaviors of the Vietnamese people, the COVID-19 epidemic was rapidly controlled. After the lockdown measures were relaxed, the Vietnamese people continued to follow COVID-19 preventive measures, and most of them scored high on the well-being scale. However, in the absence of large-scale COVID-19 vaccine coverage, new COVID-19 waves may still appear. Together with implementing preventive measures, developing strategies to guarantee the well-being of the Vietnamese people is equally important.

## Figures and Tables

**Figure 1 ijerph-18-04737-f001:**
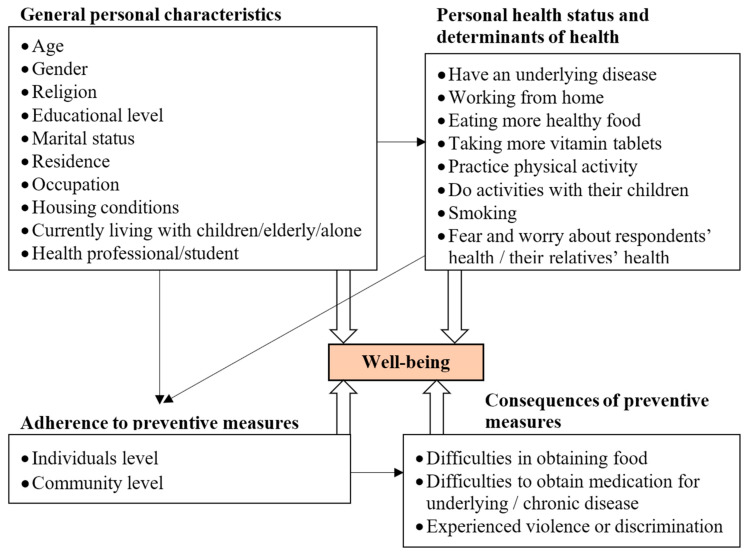
Conceptual framework of the factors associated with well-being during the COVID-19 pandemic.

**Table 1 ijerph-18-04737-t001:** COVID-19 domestic infection cases in Vietnam by the waves of outbreaks.

Stages of COVID-19 Outbreak	Total Cases	Total Deaths
1st wave (23 January 2020–16 April 2020) [5]	140	0
2nd wave (25 July 2020–2 September 2020) [6]	551	35
3rd wave (28 January 2021–18 March 2021) [7]	908	0

Source: Vietnam Ministry of Health.

**Table 2 ijerph-18-04737-t002:** Characteristics of the study participants (*n* = 1922).

Characteristic		*n*	%
Gender	Male	587	30.5
	Female	1332	69.3
	Other	3	0.2
Age group	<40 years	1590	82.7
	≥40 years	332	17.3
Adheres to a religion	Yes	417	21.7
	No	1505	78.3
Highest educational level	High school and lower	541	28.1
	University and higher	1381	71.9
Marital status	Married	886	46.1
	Other	1036	53.9
Place of residence	Municipalities	932	48.5
	Smaller urban or rural areas	990	51.5
Occupation	Student	412	21.4
	Government staff	706	36.7
	Private enterprise or self-employed	715	37.2
	Unemployed	89	4.6
Health professional or student in the health sector?	Yes	1696	88.2
	No	226	11.8
Urban/Rural or Semi-Rural residence	Urban	1139	59.3
	Sub-urban/Rural	783	40.7
Housing conditions	Private house or apartment	1697	88.3
	Renting house/room	225	11.7
Currently living:	Alone	136	7.1
	With children	1376	71.6
	With elderly persons	286	14.9
Smoking	Yes	118	6.1
	No	1804	93.9
Eating more healthy food	Yes	1699	88.4
	No	223	11.6
Taking more vitamin tablets	Yes	1206	62.7
	No	716	37.3
Have an underlying disease	Yes	135	7.0
	No	1787	93.0

**Table 3 ijerph-18-04737-t003:** Distribution of the educational level/occupation of the participants by gender (*n* = 1919).

Educational Level and Occupation		Gender *n* (%)		*p*-Value
		Male (*n* = 587)	Female (*n* = 1332)	
Highest educational level	High school and lower	138 (23.5)	403 (30.3)	0.002
	University and higher	449 (76.5)	929 (69.7)	
Occupation	Student	112 (19.1)	299 (22.4)	<0.001
	Government staff	273 (46.5)	433 (32.5)	
	Private enterprise or self-employed	169 (28.8)	544 (40.8)	
	Unemployed	33 (5.6)	56 (4.2)	

**Table 4 ijerph-18-04737-t004:** Consequences of the COVID-19 pandemic on people’s lives (*n* = 1922).

Characteristic		*N* (%)
Fear and worry about respondents’ health	Moderate or high	406 (21.1)
	None or minimal	1516 (78.9)
Fear and worry about their relatives’ health	Moderate or high	517 (26.9)
	None or minimal	1405 (73.1)
Difficulties in obtaining food	Yes	97 (5.0)
	No	1825 (95.0)
Difficulties to obtain medication for underlying disease (*n* = 135)	Yes	9 (6.7)
	No	126 (93.3)
Working from home	Yes	586 (30.5)
	No	1336 (69.5)
Experienced violence or discrimination	Yes	6 (0.3)
	No	1916 (99.7)
Physical exercise	Yes	1675 (87.1)
	No	247 (12.9)
Type of physical exercise (*n* = 1675)	Indoor, with music	589 (35.2)
	Indoor, with online video	169 (10.1)
	Outdoor	1247 (74.4)
Activities with their children (*n* = 1376)	Yes	1105 (80.3)
	No	271 (19.7)
Type of activities with their children (*n* = 1105)	Tell a story, talk about something they like, read a book, or share pictures	570 (51.6)
	Taking a walk around the house or in the street	430 (38.9)
	Doing exercises together while listening to their favorite music	214 (19.4)
	Doing a house chore together while having fun	603 (54.6)
	Getting help with their school work	444 (40.2)

**Table 5 ijerph-18-04737-t005:** Adherence to personal COVID-19 preventive measures.

Measures	March 31 to April 6 *N* = 2175 [10]	April 21 to 25 *N* = 1922
	*N* (%)	*N* (%)
Follow the 1.5–2 m physical distance rule	1919 (88.2)	1809 (94.1)
Face mask use when outdoor	2165 (99.5)	1921 (99.9)
Cover mouth and nose when coughing/sneezing	2065 (94.9)	1879 (97.8)
Usually wash/disinfect hands immediately after coughing/sneezing	1813 (83.4)	1693 (88.1)
Wash hands regularly with water and soap during the day	2119 (97.4)	1899 (98.8)
Use hand sanitizer/gel regularly during the day	1767 (81.2)	1661 (86.4)
Body temperature check at least twice a week	980 (45.1)	1075 (55.9)
Avoid touching my face, eyes, nose and mouth with my hands	1852 (85.1)	1735 (90.3)
Disinfect phone when I get home	1047 (48.1)	1129 (58.7)

**Table 6 ijerph-18-04737-t006:** Adherence to community COVID-19 preventive measures in the last seven days.

Measures	March 31 to April 6 *N* = 2175 [10]	April 21 to 25 *N* = 1922
	*N* (%)	*N* (%)
Avoided meeting or gathering with more than 10 persons	1791 (82.3)	1683 (87.6)
Avoided going to a restaurant, bar, or club	2147 (98.7)	1914 (99.6)
Avoided attending a funeral	2117 (97.3)	1874 (97.5)
Avoided going to a religious gathering	2160 (99.3)	1918 (99.8)
Avoided going to a public gym	2157 (99.2)	1917 (99.7)
Avoided going to a beauty parlor, massages, spa, hairdresser, or nail studio	2121 (97.5)	1872 (97.4)
Avoided being in a vehicle or bus with more than 5 persons	2079 (95.6)	1901 (98.9)
Avoided using common plates/spoons when eating with family	1137 (52.3)	1158 (60.2)
Avoided using common plates/spoons when eating with strangers	1986 (91.3)	1791 (93.2)
Avoided going to a market	950 (43.7)	843 (43.9)
Did not travel outside my city	2162 (99.4)	1916 (99.7)

**Table 7 ijerph-18-04737-t007:** The mean scores of each item of the WHO-5 well-being scale, and the overall well-being score (*n* = 1922).

Items and Overall Well-Being Scores	Mean ± SD	Min–Max
I have felt cheerful in good spirits	3.64 ± 1.05	0–5
I have felt calm and relaxed	3.59 ± 1.07	0–5
I have felt active and vigorous	3.34 ± 1.19	0–5
I woke up feeling fresh and rested	3.50 ± 1.17	0–5
My daily life has been filled with things that interest me	3.28 ± 1.23	0–5
Overall well-being score	17.35 ± 4.97	0–25

**Table 8 ijerph-18-04737-t008:** Regression model investigating factors associated with poor well-being during the COVID-19 pandemic *.

Co-Variates	Odds Ratio (95% Confidence Interval)	*p*-Value
Respondents aged under 40 years	1.97 (1.31–2.97)	0.001
Gender: Male	0.72 (0.52–0.98)	0.039
Adherence to the COVID-19 preventive measures	0.87 (0.81–0.93)	<0.001
Working from home	0.73 (0.55–0.98)	0.035
Physical activity during the epidemic	0.61 (0.43–0.86)	0.005
Fear and worry about their relatives’ health	2.38 (1.84–3.08)	<0.001
Eating more healthy food	0.61 (0.43–0.86)	0.005
Smoking	1.89 (1.10–3.24)	0.022

* Multiple logistic regression model was used for the analysis.

**Table 9 ijerph-18-04737-t009:** Distribution of poor well-being by the period of lockdown (*n* = 1922).

Period of Lockdown	Poor Well-Being *n* (%)		*p*-Value
	Yes (*n* = 310)	No (*n* = 1612)	
During lockdown (April 21 to 22)	204 (16.4)	1037 (83.6)	0.619
After lockdown (April 23–25)	106 (15.6)	575 (84.4)	

## Data Availability

All responses were anonymous and securely stored in a password-protected server in Belgium. The datasets generated and/or analyzed during the current study are available from the corresponding author upon reasonable request.

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
