# Peer review of "Well-Being of Healthcare Workers and the General Public during the COVID-19 Pandemic in Vietnam: An Online Survey"

_ijerph, 2021, doi:10.3390/ijerph18094737_

Round 1
Reviewer 1 Report
The title needs more information, specifically that this was a snowball survey launched in a medical school.
The study procedure section needs more information about the site of the study, the sample frame, and the sampling procedure.
The discussion needs to explicitly include the key disadvantages of the snow ball survey methodology, which include.
- Community bias: The first participants will have a strong impact on the subsequent sample.
- Non-random: Snowball sampling contravenes many of the assumptions supporting conventional notions of random selection and representativeness
- Unknown sampling population size: There is no way to know the total size of the overall population
- Anchoring: Another disadvantage of snowball sampling is the lack of definite knowledge as to whether or not the sample is an accurate reading of the target population. By targeting only a few select people, it is not always indicative of the actual trends within the result group.
.
Overall, the paper reads as if it is intended to reflect on the Vietnamese population as a whole, but since it was launched in a medical school, that can hardly be considered to be representative of the general population. I think therefore that if this paper is to be published it should be much more explicit about the sample in the title, in the methods and then in the discussion.
Author Response
Response to Reviewer 1 Comments
Thank you for the very valid comments. We have responses to your comments as follows:
Point 1:
The title needs more information, specifically that this was a snowball survey launched in a medical school. The study procedure section needs more information about the site of the study, the sample frame, and the sampling procedure.
Response 1:
The new title is now: “Well-being of healthcare workers and the general public during the COVID-19 pandemic in Vietnam: an online survey”
We also added in the abstract “The survey was launched by the Hue University of Medicine and Pharmacy… 88.2% health professionals or students in the health sector.” In the study procedures, we added: “The survey was launched by the Institute for Community Health Research, Hue University of Medicine and Pharmacy, Vietnam”
Point 2:
The discussion needs to explicitly include the key disadvantages of the snow ball survey methodology, which include.
1. Community bias: The first participants will have a strong impact on the subsequent sample.
2. Non-random: Snowball sampling contravenes many of the assumptions supporting conventional notions of random selection and representativeness
3. Unknown sampling population size: There is no way to know the total size of the overall population
Anchoring: Another disadvantage of snowball sampling is the lack of definite knowledge as to whether or not the sample is an accurate reading of the target population. By targeting only a few select people, it is not always indicative of the actual trends within the result group.
Response 2:
The limitation of the snowball sampling has now been added in the discussion.
See lines 235-244 on page 8.
We now state: “The key disadvantages of our snowball sampling based online survey include a community bias. Indeed, it is possible that the first participants would determine the remaining participants as they will share the link only to their networks. This was partially mitigated by sharing the survey link on several different platforms. The survey was launched in medical schools, which resulted in a large percentage of respondents being medical students and healthcare workers. Such a population is likely to be more aware and compliant with preventive measures. The non-random selection of participants, the selection bias of the online approach and the adoption of a convenient sample size rendered our study population non-representative of the general population in Vietnam.”
Point 3:
Overall, the paper reads as if it is intended to reflect on the Vietnamese population as a whole, but since it was launched in a medical school, that can hardly be considered to be representative of the general population. I think therefore that if this paper is to be published it should be much more explicit about the sample in the title, in the methods and then in the discussion
Response 3:
We agree that the participants in our study were not representative of the Vietnamese population.
We now state as a limitation of our study that “The survey also was launched in medical schools, which resulted in a large percentage of respondents being medical students and healthcare workers…”
We have now changed the title and adapted also the methods (see lines 80-83 on page 2) and discussion (see responses above)
Please see our revised manuscript in the attachment file

Reviewer 2 Report
Dear authors, thank you for this interesting paper, although I have some comments:
1.) Concerning the selection of your respondents there seems to be a strong bias towards young, well educated females in health and related services, so in how far are your results representative for the Vietnamese population, as the title of your paper says? You mention these limitations in a few short lines at the end of the paper, but perhaps this fact should already be made clearer at the beginning of the paper, when you talk about the survey and study design.
2) Another problem is that in chapter 4 you talk about the elderly population which seemed to have been more at ease in Vietnam than in other countries. But you do not provide information about the age distribution of your respondents (the category "age" is missing in table 2, but we know from your text that the average age of your respondents is 31 … so how valid is your statement about the Vietnamese elderly in the Covid crisis?
3) Tables 2-4 are very interesting and essential, but they will need a clearer layout, which should make it easier for the readers to link text and associated data.
4) As far as you literature review is concerned, you mention just a few studies e.g. from Austria and the US, but are there also other studies on Covid-19 issues in Vietnam? If yes, you could provide some additional information on the situation of the Vietnamese population in general, apart from the focus on predominantly young, female, well educated health professionals (for example, people in tourism related sectors of the economy, in ethnic minority areas etc. may have a different level of well-being than the group of respondents in your study).
Best regards, one of your reviewers
Author Response
Thank you very much for your useful comments. We have responses to your comments as follows:
Point 1:
Concerning the selection of your respondents there seems to be a strong bias towards young, well educated females in health and related services, so in how far are your results representative for the Vietnamese population, as the title of your paper says? You mention these limitations in a few short lines at the end of the paper, but perhaps this fact should already be made clearer at the beginning of the paper, when you talk about the survey and study design
Response 1:
Indeed, there was a bias towards young, well-educated females working in health and related services. This was a consequence of the way the survey was launched by the Hue University of Medicine and Pharmacy. We now mention this in the abstract, methods and discussion. We also changed the title to: “Well-being of healthcare workers and the general public during the COVID-19 pandemic in Vietnam: an online survey”
Please see in line 235-244 in page 8 and lines 8-83 on page 2
Point 2:
Another problem is that in chapter 4 you talk about the elderly population which seemed to have been more at ease in Vietnam than in other countries. But you do not provide information about the age distribution of your respondents (the category "age" is missing in table 2, but we know from your text that the average age of your respondents is 31 … so how valid is your statement about the Vietnamese elderly in the Covid crisis?
Response 2:
We agree we have insufficient information about an “elderly age group” to make any conclusions.
We now mention the number of respondents ≥ 60 years but because of the small number (27) in the multivariable analysis we used the age group ≥40.
“aged 40 years and above” instead of “elderly”
We now state in the discussion “In our survey only 27 (8.1%) of the 332 those aged 40 years and above were older or equal to 60 years. Moreover, the successful interruption of COVID-19 transmission in Vietnam at an early stage of the pandemic, and the absence of reported deaths due to COVID-19 during the first wave may have put the population older than 40 years at greater ease [10]. In addition, the risk to lose a job or decreased income during the pandemic might have been lower in persons above the age of 40 years as compared to younger people [21].
See for our update in line 200 -209, page 8.
We also re-analysed data for the findings related to this small change of respondent age. See age group classification is added in Table 2 and Table 8
Point 3:
Tables 2-4 are very interesting and essential, but they will need a clearer layout, which should make it easier for the readers to link text and associated data.
Response 3:
We improved of the layout of Table 2 to Table 4.
Point 4:
As far as you literature review is concerned, you mention just a few studies e.g. from Austria and the US, but are there also other studies on Covid-19 issues in Vietnam? If yes, you could provide some additional information on the situation of the Vietnamese population in general, apart from the focus on predominantly young, female, well educated health professionals (for example, people in tourism related sectors of the economy, in ethnic minority areas etc. may have a different level of well-being than the group of respondents in your study).
Response 4:
We have updated other similar findings related the well-being survey from Asia and Vietnam during social distancing in the COVID-19 pandemic.
We now include in the discussion “In another online survey in Vietnam during the lockdown measures, female gender, having chronic diseases and a high number of family members was associated with lower well-being and quality of life [17].
We added the reference of Tran, B.X et al (ref 17)
Please see in line 180-183 at page 7, and line 210-216 at page 8.
Please see our revised manuscript for your further references

Reviewer 3 Report
Title: “Vietnamese people’s well-being during the COVID-19 pandemic: an online survey”
The authors undertook an online survey of Vietnamese adults effect of preventative behaviour and well-being post lockdown. A previous study had already identified good preventative behaviour during lockdown. The study overall showed very little change in the preventative behaviour when compared with the previous study that was undertaken during lockdown, suggesting good compliance. However, the study population were principally medical students/healthcare workers which may bias the results as this population is likely to be more aware and compliant with preventative measures.
The following are my comments to improve the manuscript.
- The authors did not give the date or duration of the lockdown. They report that lockdown began on 1st April 2020 and the study was undertaken between 21-25 April 2020 but they do not give a date from when lockdown was lifted? This would be important to ascertain the significance of the findings.
- Consider revising/ restructuring Tables 2 and 4 as they are difficult to read and interpret.
- Since comparisons are being made with the previous study on preventative measures I suggest that the authors reference doing this in the analysis.
- There appears to be a poorly structured interpretation of the descriptive data and/or confusion of referencing the tables in the text. For example:
Lines 134-136 the authors suggest “Adherence to personal preventive measures remained high during this second survey with rates ranging from 55.9% to 99.9%. Only temperature check at least twice a week and disinfecting one’s phone were seldom reported. (Table 4).”- This is not clear from Table 4 information and the descriptive data in Table 5 suggests that adherence increased for phone disinfection from 48% to 58% between the two periods and generally across most factors measured overall adherence increased between the two time periods.
Lines 138-139 the authors suggest “Adherence to community preventive measures also remained high during this second survey with rates ranging from 43.9% to 99.7% but most people continued going regularly to a market (Table 5)”. Going to Market refers to Table 6 not Table 5 and the % that went to Market according to the descriptive figures in Table 6 remained similar in both time periods.
There are two tables labelled Table 6- causing confusion.
- I am confused about the analysis and suggest that the authors revisit this section and fully explain how they analysed the data. The authors indicate in the regression model the dependent variable is the WHO-5 wellbeing. How was this analysed? Binary Logistic regression models typically require the dependent variable to be binary- were any binary logistic modelling undertaken first to explore the characteristics of interest and if so, what were the parameters used for the WHO-5 variable? ( for example, it is known that there are differences in mental health well-being between genders so did the authors run any models for male and female separately?) Was ordinal logistic regression carried out with each component part of the WHO-5 questions as dependent variable(s)? Was age-standardisation undertaken to establish differences between older and younger populations? Was the model tested against all the characteristics listed in Table 4? Or did the authors use a reduced set of the most significant factors for inclusion in the multiple regression?
- Lines 149-151 The authors interpretation of their findings is contradictory. The authors say that “Factors associated with a high well-being score were older age, eating more healthy food, physical activity, working from home, and adherence to the COVID-19 preventive measures. In contrast, male gender, being worried about their relatives’ health, and being a smoker were all associated with poor well-being (Table 7)”. This is contradicted in the conclusion where the authors in Line 194 say that “Female gender was also associated with poorer well-being in our study” Furthermore Table7 Odds Ratio results indicate that ‘being worried about their relative’s health’, and being a smoker are the two main independent variables associated with poor well-being and gender appears not to be? I would have thought that an odds ratio less than one means they had lower odds of scoring lower on the component question and a higher OR >1 is indicative of a higher odds of scoring a lower score on the wellbeing question?
- The way missing data is handled can have a profound effect on the results of regression analyses. The authors give no indication of how missing data was handled in the analysis. Is it assumed because the survey forced individuals to answer all questions that there were no missing data? If this is the case the authors should refer to this in the analysis section. Additionally, a possible limitation of ‘forced response’ in survey questionnaires is that the participant may have a ‘reactance effect’ if they are unsure/uncertain and will just choose an answer to let them move on to the next question.
Author Response
Thanks for your very useful comments. The followings are our responses
Point 1:
Title: “Vietnamese people’s well-being during the COVID-19 pandemic: an online survey”
The authors undertook an online survey of Vietnamese adults effect of preventative behaviour and well-being post lockdown. A previous study had already identified good preventative behaviour during lockdown. The study overall showed very little change in the preventative behaviour when compared with the previous study that was undertaken during lockdown, suggesting good compliance. However, the study population were principally medical students/healthcare workers which may bias the results as this population is likely to be more aware and compliant with preventative measures
Response 1:
The population was biased towards medical personnel or students because the survey was launched from a university of Medicine and Pharmacy, thereby making the study population non-representative. We now mention this in the limitations.
See our modification in line 80-83 at page 2.
We now state in the discussion “The survey was launched in medical schools, which resulted in a large percentage of respondents being medical students and healthcare workers. Such a population is likely to be more aware and compliant with preventative measures. “
Please see line 239-244 on page 8.
Point 2:
The authors did not give the date or duration of the lockdown. They report that lockdown began on 1st April 2020 and the study was undertaken between 21-25 April 2020 but they do not give a date from when lockdown was lifted? This would be important to ascertain the significance of the findings.
Response 2:
The national lockdown was effective from 1-22 April, 2020
The date of lockdown was lifted April 23 2020. This has now been clearly mentioned in the manuscript.
We added this additional information in lines 42-44 at page 1 and 2 and Lines 70-73 at page 2.
Point 3:
Consider revising/ restructuring Tables 2 and 4 as they are difficult to read and interpret.
Response 3:
We added the row layout to make table 2 and 4 clearer.
Point 4:
Since comparisons are being made with the previous study on preventative measures I suggest that the authors reference doing this in the analysis.
Response 4:
We have supplemented the reference from our previous study on the COVID-19 prevention measures.
Please see line 96-99 at page 3.
Point 5:
There appears to be a poorly structured interpretation of the descriptive data and/or confusion of referencing the tables in the text. For example:
Lines 134-136 the authors suggest “Adherence to personal preventive measures remained high during this second survey with rates ranging from 55.9% to 99.9%. Only temperature check at least twice a week and disinfecting one’s phone were seldom reported. (Table 4).”- This is not clear from Table 4 information and the descriptive data in Table 5 suggests that adherence increased for phone disinfection from 48% to 58% between the two periods and generally across most factors measured overall adherence increased between the two time periods.
Lines 138-139 the authors suggest “Adherence to community preventive measures also remained high during this second survey with rates ranging from 43.9% to 99.7% but most people continued going regularly to a market (Table 5)”. Going to Market refers to Table 6 not Table 5 and the % that went to Market according to the descriptive figures in Table 6 remained similar in both time periods.
There are two tables labelled Table 6- causing confusion.
Response 5:
We have corrected the table numbers and the descriptive text about the table.
See our modification in Results section in line 116-163 in page 4-7.
Point 6:
I am confused about the analysis and suggest that the authors revisit this section and fully explain how they analysed the data. The authors indicate in the regression model the dependent variable is the WHO-5 wellbeing. How was this analysed? Binary Logistic regression models typically require the dependent variable to be binary- were any binary logistic modelling undertaken first to explore the characteristics of interest and if so, what were the parameters used for the WHO-5 variable? (for example, it is known that there are differences in mental health well-being between genders so did the authors run any models for male and female separately?) Was ordinal logistic regression carried out with each component part of the WHO-5 questions as dependent variable(s)? Was age-standardisation undertaken to establish differences between older and younger populations? Was the model tested against all the characteristics listed in Table 4? Or did the authors use a reduced set of the most significant factors for inclusion in the multiple regression?
Response 6:
We used a multiple binary logistic regression model to identify factors associated with poor well-being during COVID-19 pandemic.
Dependent variable: poor well-being (binary variable). WHO Five Well-Being Index (WHO-5) was used to indicate poor well-being if the score below 13. Quantitative data of the WHO-5 is only used to descriptive analysis. We do not use ordinal logistic regression on this article.
Covariates: All factors available in the conceptual framework (Figure 1) were used to find factors associated with the poor well-being
• First, age and gender were included in the model to adjust for other covariates.
• After that, all factors were kept in the model if they had a statistically significant relationship to poor well-being at p<0.05.
Please see our update in Table 2 to 4.
Point 7:
Lines 149-151 The authors interpretation of their findings is contradictory. The authors say that “Factors associated with a high well-being score were older age, eating more healthy food, physical activity, working from home, and adherence to the COVID-19 preventive measures. In contrast, male gender, being worried about their relatives’ health, and being a smoker were all associated with poor well-being (Table 7)”. This is contradicted in the conclusion where the authors in Line 194 say that “Female gender was also associated with poorer well-being in our study” Furthermore Table7 Odds Ratio results indicate that ‘being worried about their relative’s health’, and being a smoker are the two main independent variables associated with poor well-being and gender appears not to be? I would have thought that an odds ratio less than one means they had lower odds of scoring lower on the component question and a higher OR >1 is indicative of a higher odds of scoring a lower score on the wellbeing question?
Response 7:
Thanks for your valuable comment.
We recognize that there is confusion about the interpretation of gender in the multivariate logistics regression model. Male gender was associated with high well-being score. This mistake is corrected in line 157-160 in page 7.
Point 8:
The way missing data is handled can have a profound effect on the results of regression analyses. The authors give no indication of how missing data was handled in the analysis. Is it assumed because the survey forced individuals to answer all questions that there were no missing data? If this is the case the authors should refer to this in the analysis section. Additionally, a possible limitation of ‘forced response’ in survey questionnaires is that the participant may have a ‘reactance effect’ if they are unsure/uncertain and will just choose an answer to let them move on to the next question.
Response 8:
Thank you for your question. As we mentioned in the first paragraph of study procedures, we have designed a survey questionnaire requiring all questions were mandatory such that no missing data were recorded. Besides, the questionnaire was designed in such a way that it was easy for participants to read and answers all questions. The survey was voluntary so participants will feel no pressure to answer the questions. If they do not want to respond to the survey, they can stop the interview at any time.

Round 2
Reviewer 2 Report
Dear authors,
thank you for your revised version, I have no more requests.
Author Response
Thanks Reviewer 2 for your previous comment
Reviewer 3 Report
I would like to thank the authors for their comprehensive address of the issues I raised and congratulate them on their efforts. The only additional issue I would suggest is for the authors to consider mentioning the following:
- The period of this study overlaps pre and post lockdown but the stated objective of the study is to measure behaviour when lockdown measures were relaxed. For example: The study was undertaken between April 21st to 25th but lockdown was relaxed on April 23rd therefore respondents to the survey on April 21-22nd when lockdown is still in place- their behaviour may be influenced more by the lockdown requirements, while those responding from April 23-25th when lockdown was relaxed may have taken a different behavioural approach, (e.g. maybe their behaviour relaxed?) was this accounted for in the analysis? If it were not taken into account, I would suggest that the authors acknowledged this as a possible limitation.
Author Response
Dear Reviewer,
Once again, thank you very much for your additional useful comment. The following is our response upon your comment:
Reviewer 3
The period of this study overlaps pre and post lockdown but the stated objective of the study is to measure behaviour when lockdown measures were relaxed. For example: The study was undertaken between April 21st to 25th but lockdown was relaxed on April 23rd therefore respondents to the survey on April 21-22nd when lockdown is still in place- their behaviour may be influenced more by the lockdown requirements, while those responding from April 23-25th when lockdown was relaxed may have taken a different behavioural approach, (e.g. maybe their behaviour relaxed?) was this accounted for in the analysis? If it were not taken into account, I would suggest that the authors acknowledged this as a possible limitation
Response to reviewer
After our re analysis of data set, there is no difference about poor well-being between during and after social distancing based on the participation' response date. A similar proportion of poor wellbeing was reported by those responding April 21-22nd (lockdown period) (16.4%) and those responding April 23-25th (when the lockdown measures were relaxed) (15.6%) (Table 9).
Please see Lines 165-170, page 7 in the manuscript was already updated upon your comment.
